# Study on the Surface Settlement of an Overlying Soft Soil Layer under the Action of an Earthquake at a Subway Tunnel Engineering Site

Qiongyi Wang [1], Liang Zou [2], Yungang Niu [3], Fenghai Ma [1,3,*], Shasha Lu [4,*] and Zhideyi Fu [5]

[1] School of Mechanics and Engineering, Liaoning Technical University, Fuxin 123000, China; 47211011@stu.lntu.edu.cn
[2] Shenzhen Dasheng Surveying Technology Co., Shenzhen 518000, China
[3] College of Architecture and Engineering, Dalian University, Dalian 116622, China
[4] School of Civil Engineering, Liaoning Technical University, Fuxin 123000, China
[5] College of Physical Science and Technology, Dalian University, Dalian 116622, China
* Correspondence: mfh@dlu.edu.cn (F.M.); lss@dlu.edu.cn (S.L.)

**Abstract:** In order to study the surface settlement characteristics of an overlying soft soil layer in a subway tunnel under seismic loading, the ABAQUS finite-element analysis software was utilized. Taking the construction of Dalian Metro Line 5, which is located in seismic intensity zone VII and has an overlying soft soil layer, as the engineering background, El-Centro and Kobe waves representing class II site seismic waves, as well as an artificial seismic wave with an exceedance probability of 10%, were inputted into the analysis. The settlement characteristics of the ground surface at the construction site of the subway tunnel under the three different seismic waves were investigated, and their behaviors were theoretically analyzed. Then, the surface settlement law with respect to the tunnel roof was studied based on El-Centro wave and soft soil parameters, and a sensitivity analysis of soft soil parameters of surface settlement of the tunnel roof was carried out by the orthogonal test method. The results show that under earthquake action, the settlement of the strata within a certain range of the tunnel roof was significantly greater than that of the surrounding strata, forming a settlement trough with a width of about 8 to 20 m. The width of the settlement trough under the El-Centro wave was the largest, about 19.6 m, surpassing that of the artificially synthesized seismic waves with a probability of 10%, which was about 15.6 m, while the width of the settlement trough under the Kobe wave was the smallest, about 8.5 m. The ground surface within a range of about 20 m above the tunnel roof was most strongly affected by the seismic waves and the special lithology of the overlying soft soil layer, and the settlement was the largest. The settlement law of the settlement trough in the overlying strata of the tunnel conformed to the Peck formula. Increasing the elastic modulus of the silty soil can reduce the settlement of the ground surface above the tunnel roof; increasing the Poisson's ratio of the silty soil will increase the settlement of the ground surface above the tunnel roof; increasing the cohesive force of the silty soil to 20 kPa will basically stabilize the settlement of the ground surface above the tunnel roof; and increasing the internal friction angle of the silty soil will basically not change the settlement of the ground surface above the tunnel roof. The sensitivities of the soft soil parameters to the settlement of the ground surface above the tunnel roof were in the order of the Poisson's ratio, the elastic modulus, the cohesive force, and the internal friction angle. Therefore, the research findings of this paper provide scientific support for the problem of surface settlement of the overlying soft soil layer in subway tunnel engineering sites under earthquake action. In addition, these research findings have important theoretical value and engineering application significance, especially in the field of sustainability.

**Keywords:** overlying soft soil layer; subway tunnel; seismic wave; ground settlement; sensitivity

## 1. Introduction

Soft soil, as a type of special soil, has poor seismic performance, which causes serious impacts in engineering construction [1,2]. With the increasing number of subway construction projects in China, many subway lines are located above soft soil layers, which poses challenges for the seismic design of subway construction projects. Earthquakes, as a common type of geological hazard, have a tremendous impact on soft soil areas. The presence of soft soil causes uneven settlement and increased longitudinal curvature of tunnels located in the seismic subsidence area, seriously affecting their normal use [3–6]. According to reports on examples of tunnel damage caused by earthquakes [7], in 1923, the Kanto earthquake, with a magnitude of 7.9 in Japan, destroyed 150 tunnels, producing cracks in the tunnel lining and sinking of the arch. In 1952, the 7.2-magnitude earthquake in California, USA, severely damaged four tunnels of the South Pacific Railway, causing a range of phenomena, such as cracks and lining fractures and peeling on the tunnel surfaces [8]. In 1975, an earthquake occurred in Haicheng City, China. The tunnel site was located in a soft soil layer, and under the seismic effect, the tunnel experienced significant settlement, affecting normal operations. The experience of previous earthquakes has shown that the geological conditions of tunnel sites have an extremely important impact on seismic damage, but progress in research in this area has not been significant in recent years.

The main characteristics of earthquake activity in China are frequent earthquakes of high intensity and with shallow epicenters, making the impact of earthquake disasters in China very severe. Using the three-dimensional finite-element method and the equivalent variable stiffness method to simulate the hardening of grouting materials, Fang et al. [9] calculated and analyzed the effects of cross-tunnel shield construction on the settlement of existing tunnels under different construction parameters, such as shield thrust, tail grouting, and jack pushing force, as well as different tunnel crossing angles. Zhang et al. [10], based on a subway engineering project in Hangzhou, analyzed the settlement effects of the cut-off pile foundation of shield tunneling under an electric power tunnel, combining three-dimensional numerical simulation and measured settlement data, and conducted a numerical analysis of construction parameters, such as the face pressure and synchronous grouting pressure. Moore and Guan [11] used the continuous reflection method to study the response of double-hole tunnels in semi-infinite media under seismic action, obtained the tunnel response formula under different incident waves, and compared it with a two-dimensional analysis. Hamid [12] studied the factors affecting tunnel settlement through numerical simulation, and the results showed that the fault-zone thickness and tunnel strata properties significantly affected the maximum settlement of the tunnel, and the fault affected the longitudinal settlement of the ground within 25 m on both sides of the fault center. Yang et al. [13] studied the soft soil foundation in Shanghai, simulated the vibration response of the soft soil foundation under seismic action, and preliminarily summarized the dynamic characteristics of the soft soil in the area, accumulating data and experience for future research. Milind Patil et al. [14] studied the seismic response of shallow-buried tunnels under soft soil foundations and obtained the seismic response law of tunnels under different working conditions of soft soil foundations. Samanta et al. [15] studied the structural response of a 15-story building using SAP2000 for modeling and performing nonlinear time–history analyses. They used four different ground-motion modification methods to analyze the response of the structure to both short- and long-term earthquake motions.

Xia et al. [16] used the soft soil in a certain area of Tianjin Binhai as a research object. They applied three levels of earthquake waves, strong, medium, and weak, to the soft soil site. The experimental results showed that the seismic motion of the soft soil was very little affected by the intensity of the earthquake waves. However, when the dynamic shear modulus of the soft soil changed, the dynamic characteristics of the soft soil site changed significantly. Zuo et al. [17] conducted vibration table tests based on soft clay conditions, focusing on the vibration responses of subway stations under seismic action in

far-field and near-field sites, and analyzed the changes in dynamic response parameters of underground structures.

The above-mentioned research was mainly conducted from the perspectives of statics and dynamics to study the characteristics of soft soils and the response of metro tunnels passing through soft soil layers. However, there are few studies on the problem of surface settlement of the overlying soft soil layers at construction sites of metro tunnels under seismic action. Therefore, as shown in Figure 1, the Dalian Metro Line 5 Section 04, from the Railway Station to Suoyu Bay South Station, with an overlying soft soil layer, was taken as a case study. Using ABAQUS finite-element analysis software, the surface settlement characteristics of the overlying soft soil layer at the subway tunnel construction site under seismic action were deeply explored and theoretically analyzed. Then, based on the El-Centro wave and soft soil parameters, the settlement characteristics of the tunnel roof surface were studied. Finally, a sensitivity analysis of soft soil parameters related to the settlement of the tunnel roof surface was conducted using the orthogonal test method.

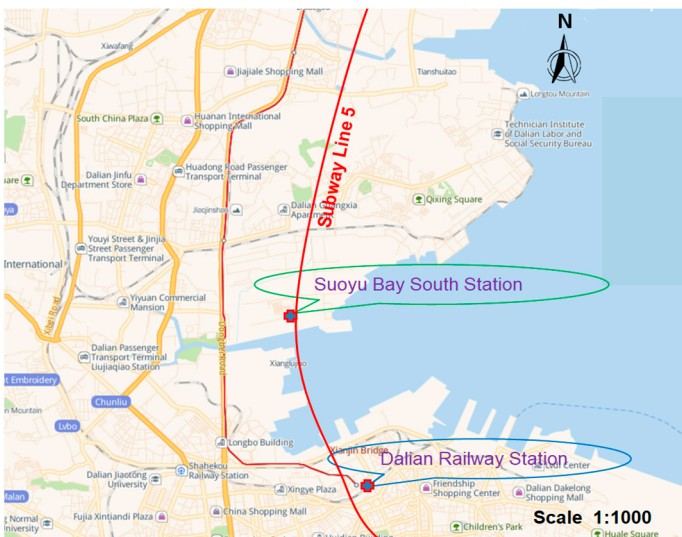

**Figure 1.** Tunnel for the Dalian Metro Line 5 Section 04 between the Railway Station and Souyuan Bay South Station.

## 2. Numerical Model of the Metro Tunnel with an Overlying Soft Soil Layer

### 2.1. Convergence–Confinement Method Based on Elastic–Plastic Analysis

Elastic–plastic analysis, also known as elastic limit equilibrium analysis, is based on Coulomb's theory and assumes that a rock mass does not change after failure. An ideal elastic–plastic model can be used to well describe the deformation and failure characteristics of underground chambers and surrounding rocks. Through the analysis of Fenner and H. Kastner, the characteristic curve of the surrounding rock of the tunnel was obtained. The study shows that under the condition of low support resistance, the self-supporting force of the surrounding rock can be used to achieve stability [18]. This achievement has been widely promoted and adopted by the New Austrian Method and the convergence–confinement method. With the continuous improvement of tunnel construction monitoring levels, the convergence–confinement method theory has gradually been applied to practical tunnel engineering, mainly reflected in the design and stability analysis of tunnels. According to the measured deformation of the tunnel section during construction, the excavation of a tunnel is regarded as a process of re-distribution of the stress of the surrounding rock [19]. Therefore, it is particularly important to scientifically and reasonably simplify the tunnel construction model, especially the simulation of stress release of the surrounding rock during excavation.

Using the convergence–confinement method, the nodal forces of the excavation face were studied, and the nodal forces adapted to the initial stress were obtained. Based

on this, nodal forces were applied to each node, and the amplitude of the nodal forces decreased over time. When it drops to a certain level, the lining unit is activated, and then the remaining load decays. Based on this, an orthogonal experimental model with convergence–confinement focusing on the stress release of the surrounding rock during excavation was established using ABAQUS, which was used to systematically examine the settlement of the top surface of the soft soil layer under the seismic action of various parameters of the soft soil layer (elastic modulus, cohesive force, Poisson's ratio, and internal friction angle).

### 2.2. Calculation of Geometric Dimensions and Parameters of the Model

The numerical simulation performed in this study was based on the construction of the Dalian Metro Line 5. According to the survey report and design information, the stratigraphic layers are described as follows: ① plain fill soil: layer thickness of 5.8 m; ② silty soil: layer thickness of 1.7 m; ③ pink sand: layer thickness of 3.2 m; ④ clay with gravel: layer thickness of 2.5 m; and ⑤ moderately weathered slate: layer thickness of 51.2 m. The distance from the roof of the tunnel to the ground net distance is 17.2 m. To overcome the influence of boundary conditions, the width of the tunnel is set to 3~5 times the length of the tunnel. Therefore, the model's length, width, and height are 177 m × 30 m × 64.4 m, as shown in Figure 2. The physical and mechanical parameters of the soil layers in the model were calculated and are shown in Table 1. The initial lining material of the supporting structure is C60, with a thickness of 30 cm, and the parameters of the model's initial lining are shown in Table 2.

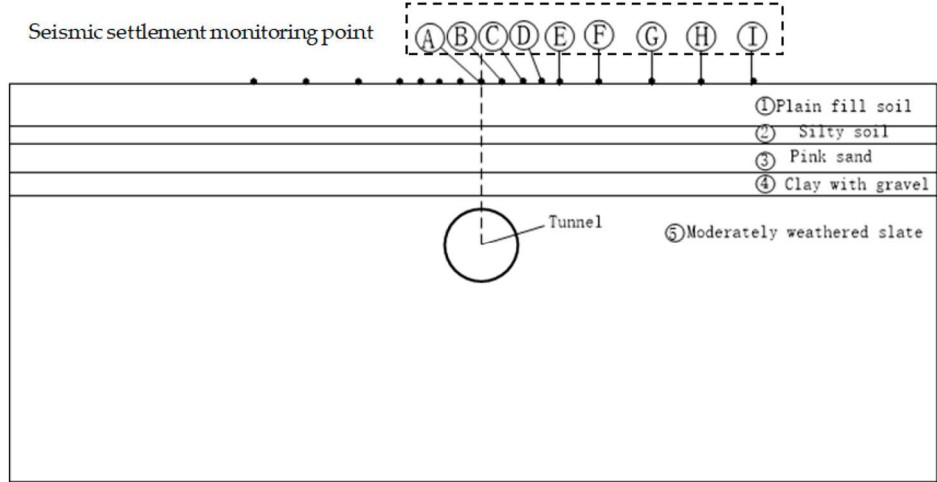

**Figure 2.** Map of soil layer distribution.

**Table 1.** Physical and mechanical parameters of soil layer.

| Stratigraphic (Genetic) | Layer Thickness/m | Density /kg·m$^{-3}$ | Elasticity Modulus *E*/MPa | Cohesion *c*/KPa | Internal Friction angle *φ*/° | Poisson's Ratio *v* |
|---|---|---|---|---|---|---|
| ① Plain fill soil | 5.8 | 1850 | 60 | 12 | 10 | 0.38 |
| ② Silty soil | 1.7 | 1800 | 30 | 15 | 3 | 0.42 |
| ③ Pink sand | 3.2 | 1900 | 200 | 5 | 29.2 | 0.20 |
| ④ Clay with gravel | 2.5 | 1950 | 280 | 30 | 25 | 0.25 |
| ⑤ Moderately weathered slate | 51.2 | 2500 | 5000 | 50 | 40 | 0.15 |

**Table 2.** Mechanical parameters of lining structure.

| Density/kg·m$^{-3}$ | Thickness/m | Elasticity Modulus *E*/Mpa | Poisson's Ratio *v* |
|---|---|---|---|
| 2450 | 0.3 | 36,500 | 0.23 |

The existence of soft soil poses a significant risk of settlement to the top of the tunnel's ground surface. In this study a three-dimensional finite-element model was established using ABAQUS. The model was symmetrically distributed along the midpoint A, and the measurement points B1~I1 on the left side of point A of the top of the tunnel's ground surface corresponded to measurement points B~I on the right side, respectively. The distance between points A and E was 3.3 m, the distance between points E and F was 6.6 m, and the distance between the remaining points was 9 m, as shown in Figure 1. To explore the impact of the parameters of the silts on the settlement of the top of the tunnel's ground surface under seismic action, the average settlement values for the top of the tunnel's ground surface at measurement points A~I and B1~I1 under three seismic waves were analyzed, with the maximum settlement not exceeding the specification value of 30 mm. The relationship between the parameters of the soft soil ($E$, $c$, $\varphi$, and $\nu$) and the maximum settlement of the top of the tunnel's ground surface was studied.

### 2.3. Model Materials and Boundary Conditions

Based on the construction of the Dalian Metro Line 5, a numerical model was established using the finite-element software ABAQUS, with a local damping coefficient of 0.1491. In the load-setting options, the displacement and rotation of the four sides of the soil model's front, rear, left, and right were constrained; in step one of the analysis, the rotation and displacement of the bottom of the soil model were constrained, and the bottom of the model was fixed. In step two of the analysis, the normal and tangential displacement of the bottom of the model were limited. To obtain the best calculation accuracy, the mapped mesh division method was used to ensure the regularity of the mesh structure. The final model is shown in Figure 3, where layers ① to ⑤ are loess, silts, fine sand, clay with gravel, and moderately weathered slate, respectively. The soil was modeled using the Mohr–Coulomb constitutive model.

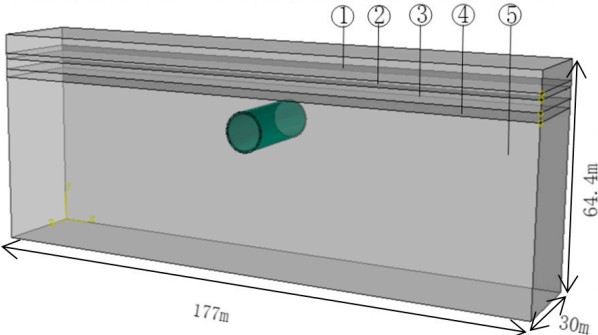

**Figure 3.** Three-dimensional finite-element analysis model.

According to previous research [20], a circular tunnel model was established in a uniform and elastic semi-infinite region, and the variations in stress and displacement were around 5% within three times the diameter range and less than 1% beyond five times the range. To overcome the influence of the boundary conditions, the width of the tunnel was set to 3~5 times the length of the tunnel. Therefore, the model had a length, width, and height of 177 m × 30 m × 64.4 m in order to overcome the influence of the boundary on the calculation results. A hybrid mesh was used, consisting of a total of 12,500 mesh elements.

### 2.4. Selection of Seismic Loads

Based on the engineering background of Dalian Metro Line 5, the seismic activity in Dalian was analyzed according to the "China Seismic Ground Motion Parameter Zonation Map" (GB 18306-2015) at a scale of 1:4 million [21], and simulated seismic inputs were obtained using a 0.1 g El-Centro wave, a 0.1 g Kobe wave, and an artificial earthquake wave with a probability of exceedance of 10% to simulate the effect of the earthquake in

Dalian. The acceleration time–history curves for the three types of earthquake waves are shown in Figure 4.

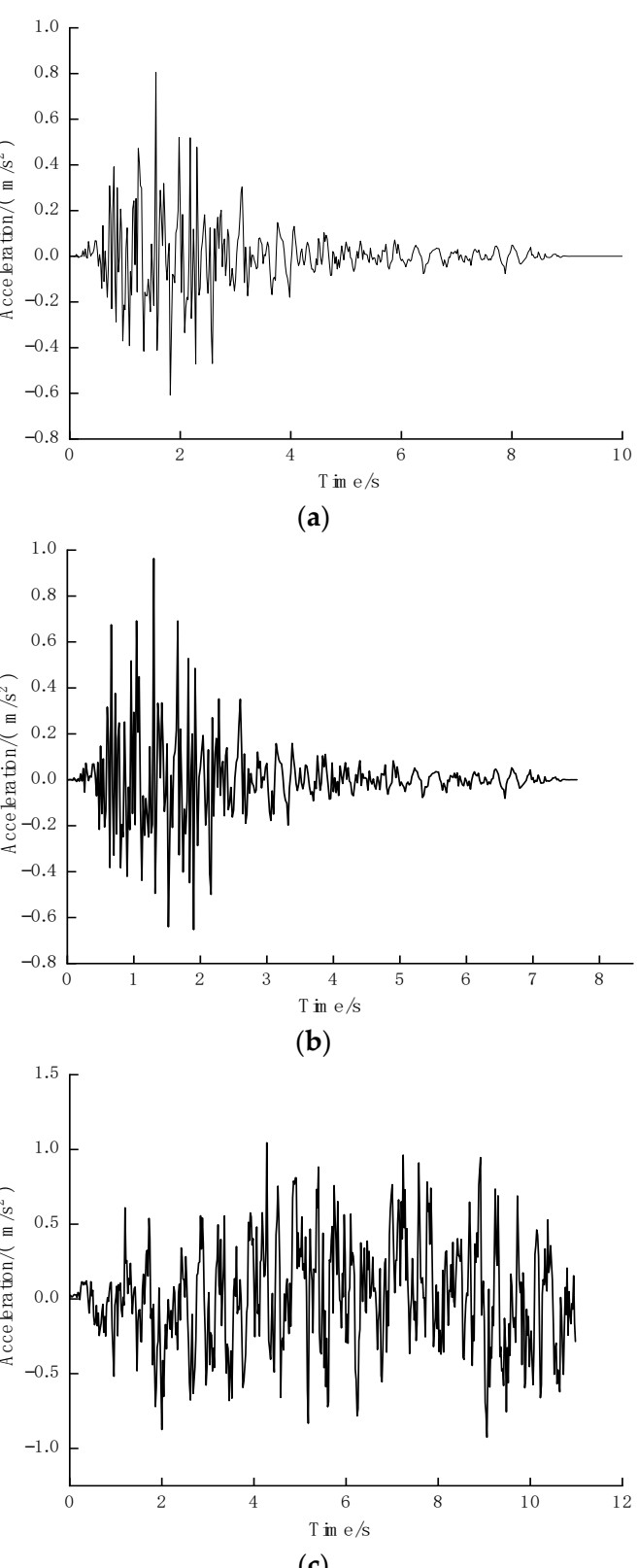

**Figure 4.** Earthquake wave acceleration time-course curves. (**a**) El-Centro wave. (**b**) Kobe wave. (**c**) Artificial seismic wave beyond a probability of 10%.

## 3. Results and Discussion

### 3.1. Settlement Deformation Law of the Tunnel Roof Strata

A displacement cloud diagram for the vertical direction of the model after the earthquake was calculated using ABAQUS is shown in Figure 5. The unit for the horizontal-direction *Y*-axis is in meters.

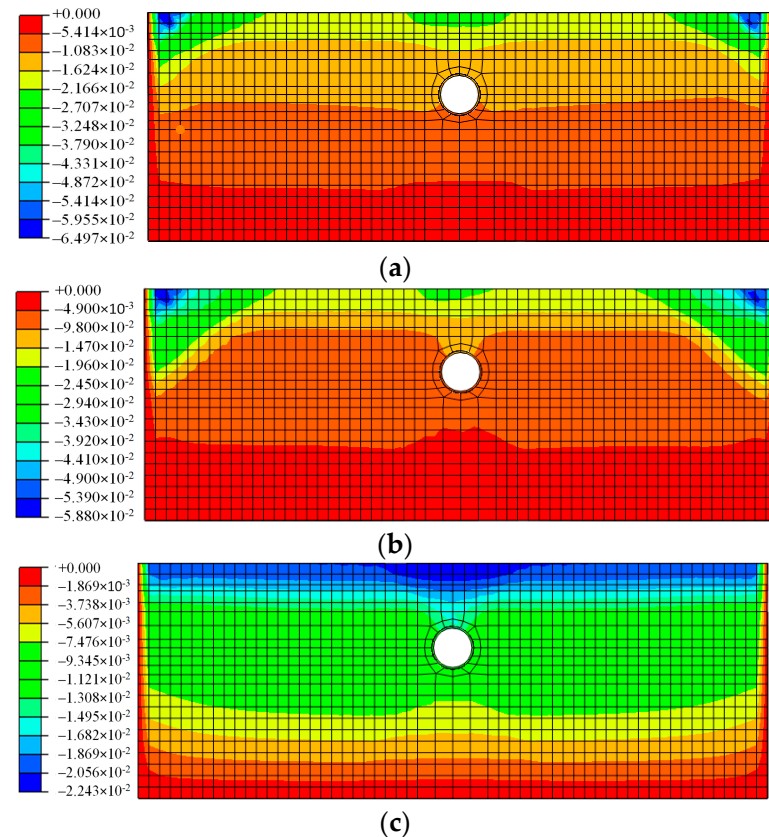

**Figure 5.** Vertical displacement of the cloud map under different seismic waves. (**a**) Artificial seismic wave beyond a probability of 10%. (**b**) El-Centro wave. (**c**) Kobe wave.

From the figure, it can be seen that there is significant differential settlement in the soft soil strata, and the strata directly above the tunnel settle significantly higher than the surrounding strata under earthquake action, forming a settlement trough, with the maximum settlement located directly above the tunnel. Comparing the results under the three different earthquake waves, it was found that the width of the settlement trough was largest under the El-Centro wave, at about 19.6 m, followed by the artificial earthquake wave with a probability of exceedance of 10%, at about 15.6 m, while the width of the settlement trough was the smallest under the Kobe earthquake wave, at about 8.5 m.

### 3.2. Analysis of Ground Settlement Theory

When an earthquake occurs, soft soil layers are easily compressed, causing uneven settlement of the ground. The pattern and amount of deformation due to uneven settlement can be provided by an earthquake department, but the data which they provide are not universal and are often only applicable to the study of ground subsidence in small seismic zoning areas, as shown in Figure 6. In analytical procedures, it is generally assumed that the ground settlement below the interval tunnel conforms to the Peck formula [22]:

$$S(x) = S_{max} \exp\left(-\frac{x^2}{2i^2}\right) = \frac{V_i}{\sqrt{2\pi}i} \exp\left(-\frac{x^2}{2i^2}\right) \tag{1}$$

where $S_{\max}$ is the maximum settlement of the ground surface at the tunnel axis position in meters, $V_i$ is the ground loss of the unit length of the tunnel in cubic meters per meter, $x$ is the distance from the tunnel axis to the calculation point in meters, and $i$ is the width coefficient of the settlement trough, which represents the distance from the center point of the settlement curve to the inflection point in meters.

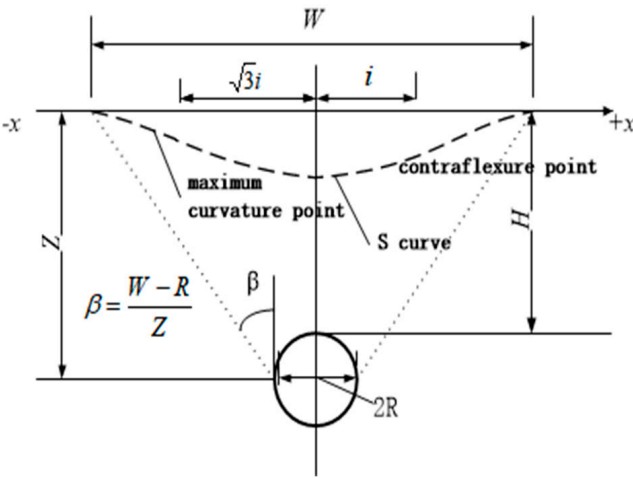

**Figure 6.** Schematic diagram of lateral prediction of surface settlement.

The distance $i$ between the center of the tunnel and the inflection point of the settlement curve is calculated using the following empirical formula:

$$i = \frac{H + R}{\sqrt{2\pi} \tan\left(45° - \frac{\phi}{2}\right)} \tag{2}$$

where $H$ is the overburden thickness, $R$ is the hydraulic radius of the tunnel, and $\varphi$ is the internal friction angle of the ground.

The maximum simulated value of ground settlement at the top of the tunnel is 21.02 mm; the maximum settlement value calculated by the formula is 21.5. Within the range of 0~40 m from the tunnel axis, the simulated value of ground settlement at the top of the tunnel is very close to the result obtained from the formula, as shown in Figure 7. Therefore, this model can reasonably reflect the impact of earthquakes on the ground settlement at the top of the tunnel.

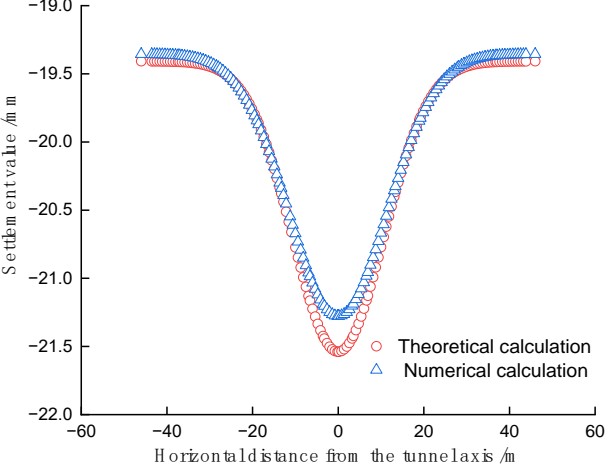

**Figure 7.** Comparison of tunnel settlement value under El-Centro wave.

## 4. Analysis of the Impact of Soft Soil Parameters on Ground Settlement at the Top of the Tunnel

### 4.1. The Impact of Elastic Modulus on Ground Settlement at the Top of the Tunnel

Under the action of earthquakes, the presence of soft soil layers has a significant impact on the ground settlement at the top of the tunnel. Based on engineering experience and using El-Centro waves, the ground settlement curves at the top of the tunnel were simulated and data were organized for elastic moduli of 20 MPa, 25 MPa, 30 MPa, 35 MPa, 40 MPa, and 50 MPa for silty clay, as shown in Figure 8. It can be seen that the ground settlement curve at the top of the tunnel reaches its maximum value at the tunnel center axis and that the maximum settlement decreases from the middle to the sides. As the elastic modulus increases, the width of the settlement trough changes less, the depth of the settlement trough decreases, and the settlement of the tunnel decreases. The influence of the elastic modulus of the soft soil on the ground settlement at the top of the tunnel under earthquake action is quite obvious. In practical engineering, it is necessary to reinforce soft soil and increase the elastic modulus of the soil to control the settlement of the tunnel roof surface.

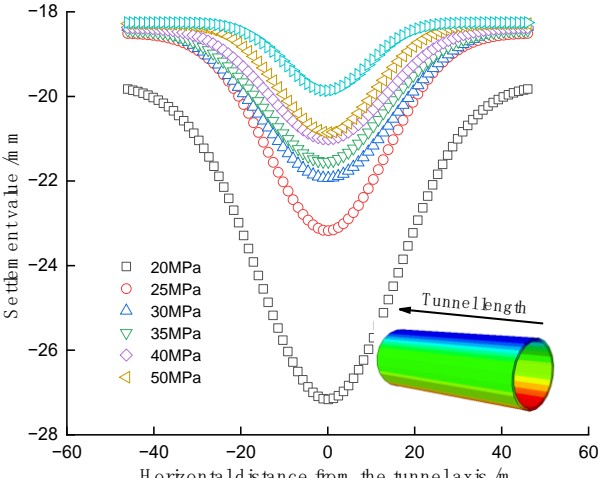

**Figure 8.** Influence of elastic modulus on surface settlement of tunnel roof.

Data processing was carried out for the elastic modulus and the maximum surface settlement of the tunnel roof in cohesive soil, and the results are shown in Figure 9. From the figure, it can be seen that under earthquake action, the elastic modulus of cohesive soil and the maximum surface settlement of the tunnel roof vary in a parabolic relationship. With the increase in the elastic modulus, the maximum surface settlement of the tunnel roof gradually decreases. This is because the increase in the elastic modulus of the soil layer leads to an increase in its stiffness index, and the deformation of the soil layer is smaller under earthquake action. The elastic modulus and maximum settlement value of cohesive soil are shown in Figure 9. For every 5 MPa increase in the elastic modulus, the corresponding maximum surface settlement of the tunnel roof decreases by about 1 mm. The fitting formula is written as follows:

$$S_x(A) = -26.20536 + 0.20424\,A - 0.0016\,A^2 \tag{3}$$

where $S_x$ is the maximum settlement value in mm and $A$ is the elastic modulus in MPa. The coefficient $R^2$ obtained from Equation (3) is 0.94465, indicating a high degree of fit. The quadratic function relationship between the maximum settlement value of the tunnel roof (mm) and the elastic modulus (MPa) is obvious, and the parabola opens downward. With the increase in the elastic modulus, the maximum settlement of the tunnel roof continuously decreases.

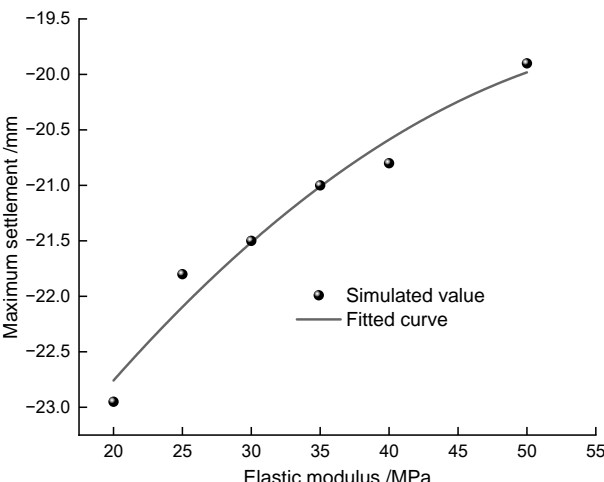

**Figure 9.** The modulus of elasticity fits the maximum settlement value to the curve.

### 4.2. The Impact of Cohesion on Ground Settlement at the Top of the Tunnel

The influence of cohesion on the surface settlement of the tunnel roof was studied by changing only the cohesion of the soft soil under earthquake conditions. The cohesion of the soft soil was set to 5 kPa, 10 kPa, 15 kPa, 20 kPa, 25 kPa, and 30 kPa. By analyzing the calculation results, the variation law of the surface settlement of the tunnel roof with the cohesion of the soft soil under El-Centro wave action was studied. The influence of cohesion on the surface settlement of the tunnel roof is shown in Figure 10.

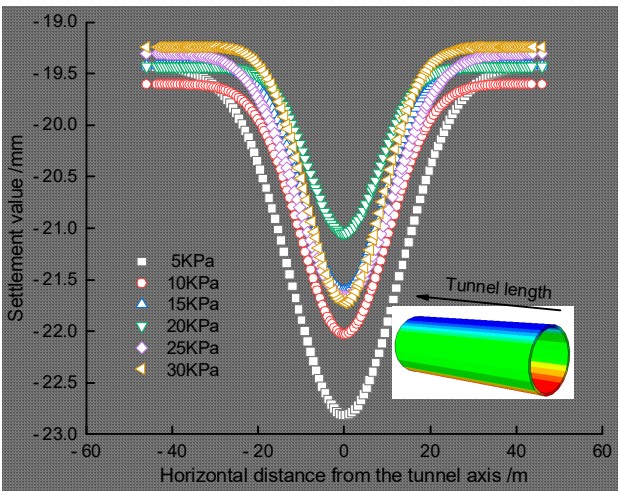

**Figure 10.** Influence of cohesion on surface settlement of tunnel roof.

From Figure 10, it can be seen that the curve for the surface settlement of the tunnel roof is symmetrical and U-shaped and that the settlement gradually tends to be horizontal along the center axis of the tunnel. With the increase in the cohesive force of the soil layer, the surface settlement of the tunnel roof continuously decreases. Under different cohesive forces, the surface settlement of the tunnel roof is larger in the middle and smaller on both sides. However, the change in settlement is relatively small. In the range of 25 kPa to 30 kPa, the surface settlement of the tunnel roof basically does not change with the increase in the cohesive force.

Data processing was carried out for the cohesive force of the silty clay and the maximum settlement value of the tunnel's top surface, and the results are shown in Figure 11. As shown in the figure, the maximum settlement value for the tunnel's top surface decreases as the cohesive force increases for cohesive forces of 5 kPa, 10 kPa, 15 kPa, and 20 kPa, and it remains stable for cohesive forces of 25 kPa and 30 kPa. Therefore, the optimal

cohesive force for controlling the maximum settlement value of the tunnel's top surface is approximately 20 kPa. The fitting formula obtained from the processed data is:

$$S_x(B) = -23.622 + 0.22083\,B - 0.00523\,B^2 \tag{4}$$

where $S_x$ is the maximum settlement value in mm and $B$ is the cohesive force in kPa.

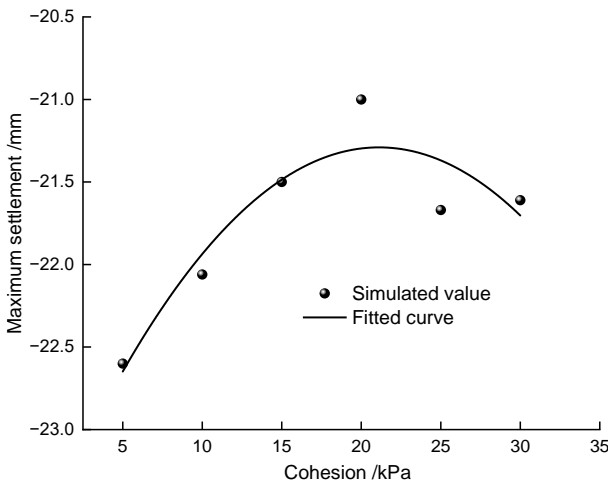

**Figure 11.** The cohesion fits the curve to the maximum settlement value.

Formula (4) is a quadratic function that represents the maximum settlement value for the tunnel's top surface under seismic action and the cohesive force of the silty clay. The determination coefficient of the fitting, $R^2$, is 0.76736, which is lower than the linear fitting of the elastic modulus.

### 4.3. The Impact of Poisson's Ratio on Ground Settlement at the Top of the Tunnel

The Poisson's ratio of soil is an important physical property index that reflects the initial density of the soil and, to some extent, its ability to resist deformation. By only changing the Poisson's ratio of the silty clay and keeping the other parameters constant, the results were calculated for Poisson's ratios of 0.30, 0.34, 0.38, 0.40, 0.42, and 0.46. The results were analyzed to study the changes in the settlement of the tunnel's top surface with the change in the Poisson's ratio of the silty clay under the action of the vertical self-weight and El-Centro wave. The simulated and processed data are shown in Figure 12.

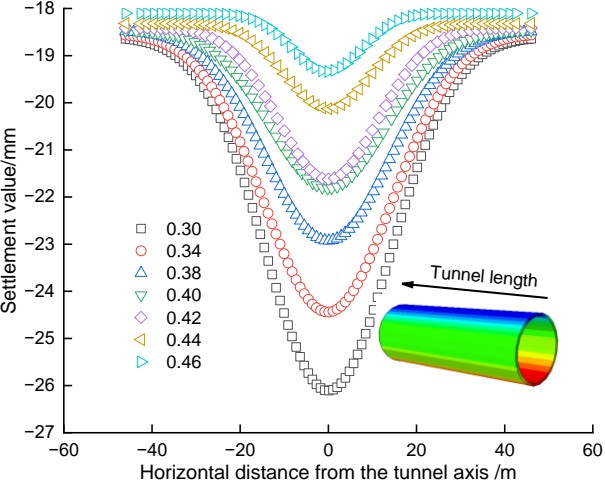

**Figure 12.** Influence of Poisson's ratio on ground settlement of tunnel roof.

It can be observed that under seismic action, the maximum settlement of the tunnel's top surface gradually decreases with the increase in the Poisson's ratio. The settlements at Poisson's ratios of 0.3 and 0.34 differ significantly, while the settlements at Poisson's ratios of 0.4 and 0.42 differ slightly. The settlement of the tunnel's top surface varies significantly for Poisson's ratios between 0.38 and 0.42, meeting the settlement requirements.

Data processing was carried out for the Poisson's ratio of the silty clay and the maximum settlement value of the tunnel's top surface, and the results are shown in Figure 13. It can be observed that the maximum settlement value for the tunnel's top surface gradually decreases as the Poisson's ratio increases. As the Poisson's ratio increases from 0.3 to 0.46, the maximum settlement value decreases sequentially, and the value changes significantly from 25.83 mm to 19.34 mm, indicating that the Poisson's ratio has a significant impact on the settlement value. The fitting formula is:

$$S_x(C) = -32.53994 + 11.21786\,C + 37.90091\,C^2 \qquad (5)$$

where $S_x$ is the maximum settlement value in mm and $C$ is the Poisson's ratio.

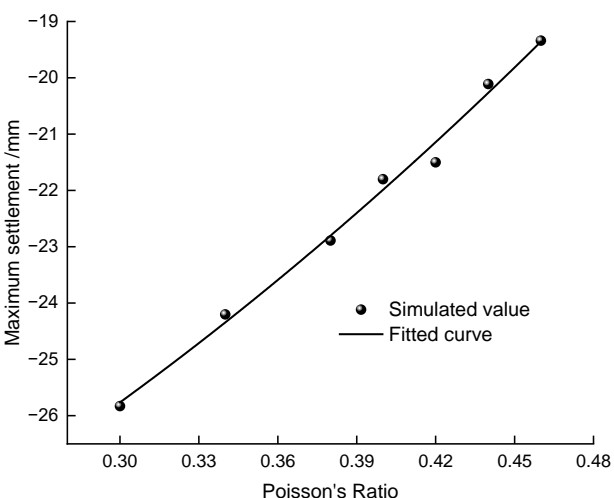

**Figure 13.** The Poisson's ratio fits the curve to the maximum settlement value.

From Formula (5), it can be derived that the Poisson's ratio and the maximum settlement of the tunnel roof surface are in a quadratic function relationship. Therefore, the optimal Poisson's ratio for controlling the maximum settlement of the tunnel roof surface in soft soil is approximately 0.46. The surface settlement value increases with the increase in the Poisson's ratio, and the determined coefficient $R^2$ of the fitting is 0.98926, which is higher than the linear fitting of the elastic modulus.

### 4.4. The Impact of Internal Friction Angle on Ground Settlement at the Top of the Tunnel

The internal friction angle is one of the parameters that affects the performance of soft soil. Keeping other parameters constant and only changing the internal friction angle of the soft soil, the situations with internal friction angles of 0°, 1°, 2°, 3°, 4°, and 5° were calculated, and the simulation results are shown in Figure 14. The results show that changing the internal friction angle of the silty soil had no significant effect on the settlement of the tunnel roof surface.

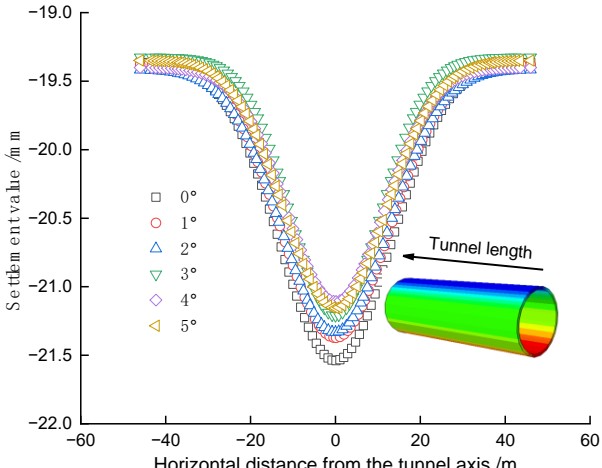

**Figure 14.** Influence of internal friction angle on surface settlement of tunnel roof.

*4.5. The Impact of Various Factors on Ground Maximum Settlement at the Top of the Tunnel*

　　The influence of various factors on the maximum settlement of the tunnel roof surface is shown in Figure 15 under seismic action. From the figure, it can be seen that when the elastic modulus of the soft soil is 20 MPa, the maximum settlement value of the tunnel roof surface is −22.96 mm; when the elastic modulus is 25 MPa, the maximum settlement value is −21.88 mm; when the elastic modulus is 50 MPa, the maximum settlement value is −19.9 mm; and when the elastic modulus of the soft soil is 30 MPa, 35 MPa, and 40 MPa, the maximum settlement does not exceed the standard value of 30 mm. The relationship between the cohesive force of the soft soil and the settlement of the tunnel roof surface is as follows: when the cohesive force is 5 KPa, the maximum settlement of the tunnel roof surface is −22.6 mm; when the cohesive force is 25 KPa, the maximum settlement is −21.67 mm; and when the cohesive force is 30 KPa, the maximum settlement is −21.61 mm. The settlement of the tunnel roof surface under soft soil cohesive forces of 15 KPa, 25 KPa, and 30 KPa varies little. When the cohesion is between 5 kPa and 20 kPa, the settlement of the tunnel roof surface tends to decrease as the cohesion increases. However, when the cohesive force increases to 20 KPa, the maximum settlement value of the tunnel roof surface stabilizes.

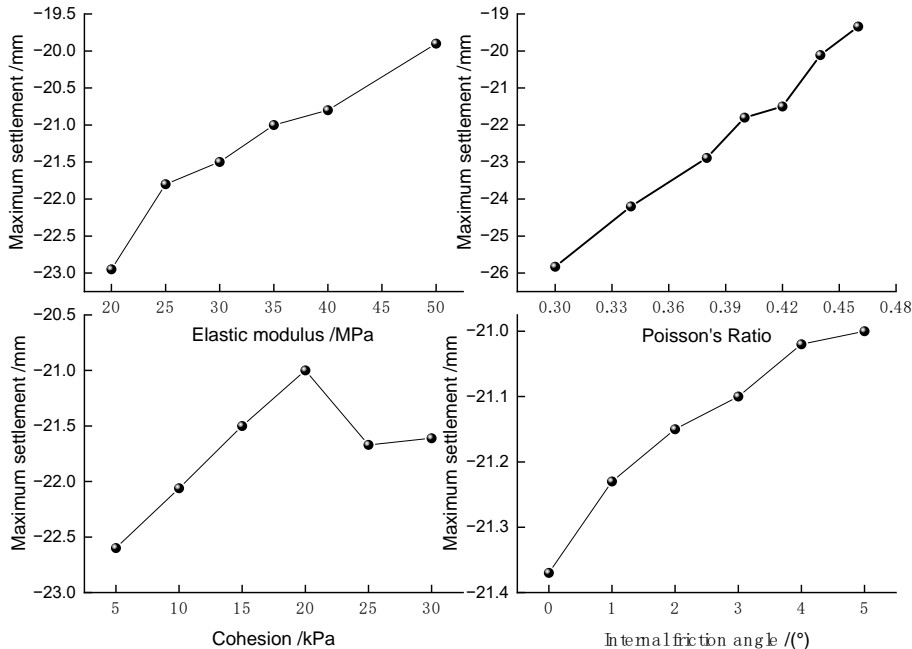

**Figure 15.** Maximum ground settlement of tunnel roof under different factors.

In summary, increasing the cohesive force of the soft soil reduces the settlement of the tunnel roof surface, and after reaching a certain value, the maximum settlement tends to stabilize. In addition, the influence of the cohesive force of the soft soil on the settlement of the tunnel roof surface is lower than that of the elastic modulus of the soil layer. Therefore, blindly increasing the cohesion of soft soil to reduce the settlement of the tunnel roof surface is not feasible. Instead, a reasonable treatment of the soft soil layer should be carried out based on the actual project. The maximum settlement of the tunnel roof surface decreases continuously with the increase in the soft soil's Poisson's ratio, indicating that the Poisson's ratio of soft soil has the greatest impact on the settlement of the tunnel roof surface. As the internal friction angle increases from 0° to 5°, the range of maximum settlement values of the tunnel roof surface changes between 21 mm and 21.4 mm, and the settlement of the tunnel roof surface does not change significantly with the variation in the internal friction angle.

## 5. Orthogonal Experiment

### 5.1. Orthogonal Experimental Design

The orthogonal experimental design uses the method of "uniform distribution and comparability" to greatly simplify the experimental scheme, thereby reflecting the advantages of orthogonal experimental design. Orthogonal experimental design can obtain more scientific results and provide an effective calculation method for solving multi-factor optimization problems [23–25]. There are many factors that affect the surface settlement of tunnel roofs under seismic action. According to the survey report for Dalian Metro Line 5 and the analysis results of Section 3, in this section of the study, the *E*, *c*, and *ν* silty clay soil parameters were selected as the three factors of the orthogonal experiment to establish a three-factor and three-level table, as shown in Table 3. The specific orthogonal experimental scheme is shown in Table 4.

**Table 3.** Factors and levels.

| Factor<br>Level | A<br>Elastic Modulus *E*/MPa | B<br>Poisson's Ratio *ν* | C<br>Cohesion *c*/kPa |
|---|---|---|---|
| 1 | 30 | 0.42 | 10 |
| 2 | 35 | 0.4 | 15 |
| 3 | 40 | 0.38 | 20 |

**Table 4.** Orthogonal tests.

| Test Number | A | B | C |
|---|---|---|---|
| 1 | 1 | 1 | 1 |
| 2 | 1 | 2 | 2 |
| 3 | 1 | 3 | 3 |
| 4 | 2 | 1 | 2 |
| 5 | 2 | 2 | 3 |
| 6 | 2 | 3 | 1 |
| 7 | 3 | 1 | 3 |
| 8 | 3 | 2 | 1 |
| 9 | 3 | 3 | 2 |

### 5.2. Orthogonal Experimental Results

In order to ensure the credibility of the experimental results and reduce the influence of unrelated engineering factors, the orthogonal table was analyzed by modulus analysis using ABAQUS software 2021 version nine times. The surface settlement curve for the tunnel roof is shown in Figure 16. The maximum settlement meets the design requirements.

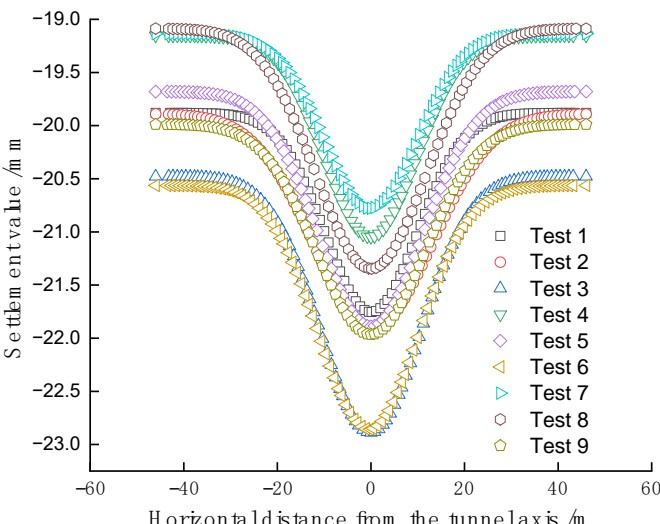

**Figure 16.** Orthogonal test settlement curve.

### 5.3. Sensitivity Analysis

The range analysis method is a simple, intuitive, and easy-to-understand algorithm. The range $R_j$ is used to analyze the influence of each factor on the evaluation index. The higher the $R_j$ value, the greater the corresponding factor's influence on the target parameter and the greater its importance.

$$R_j = \max (k_1, k_2, \ldots, k_i) - \min (k_1, k_2, \ldots, k_i) \tag{6}$$

where $k_i$ is the average value of the index corresponding to the *i*-th level and $K_i$, that is, $k_i = K_i/r$, where $r$ is the number of occurrences of any column at the same level.

By using the finite-element software ABAQUS to perform numerical simulation analysis on the orthogonal experimental scheme and taking the maximum surface settlement value for the tunnel roof in the analysis results as the evaluation index, a range analysis was conducted on the index. The specific results are shown in Table 5.

**Table 5.** Results of extreme analysis.

| Test Number | Maximum Settlement/mm | | |
| --- | --- | --- | --- |
| | **A** | **B** | **C** |
| $K_1$ | 66.26 | 63.4 | 65.61 |
| $K_2$ | 65.47 | 64.84 | 64.73 |
| $K_3$ | 63.96 | 67.45 | 65.35 |
| $k_1$ | 22.087 | 21.33 | 21.87 |
| $k_2$ | 21.823 | 21.613 | 21.577 |
| $k_3$ | 21.32 | 22.483 | 21.783 |
| $R_j$ | 0.767 | 1.153 | 0.223 |

Due to changes in the elastic modulus and the Poisson's ratio, the settlement of the tunnel's top surface underwent significant changes. Therefore, in practical engineering, the influence of soft soil on the settlement of the tunnel's top surface should be emphasized. The degree of influence of soft soil parameters on the settlement of the tunnel's top surface under earthquake loading is in the order of $v$, $E$, and $c$.

## 6. Conclusions

The aim of this study is to investigate the surface settlement of the overlying soft soil layer of subway tunnels under seismic loading. Using the Dalian Metro Line 5 as a case study, the ABAQUS finite-element analysis method was employed to simulate the

ground motion induced by an El Centro wave, a Kobe wave, and an artificial earthquake wave with 10% exceedance probability. The surface settlement was studied based on the El-Centro wave and four soft soil parameters. Orthogonal experiments and range analysis were conducted to analyze the soft soil parameters, and the main conclusions obtained were as follows:

(1) The settlement within a certain range above the tunnel is significantly greater than the surrounding layers. The maximum width of the settlement trough occurred under the influence of the El-Centro wave, followed by the synthetic artificial earthquake wave with a 10% exceedance probability, and the minimum width of the settlement trough occurred under the influence of the Kobe wave.

(2) Within a range of about 20 m of the tunnel's top surface, the seismic and special rock–soil layers of the overlying soft soil have the strongest impact on the surface settlement, and the settlement law of the overlying strata conforms to Peck's formula.

(3) The surface settlement above the tunnel is negatively correlated with the elastic modulus and cohesion of silty clay, is positively correlated with the Poisson's ratio, and is essentially unaffected by the internal friction angle.

(4) The sensitivity of soft soil parameters to the settlement of the tunnel's top surface under earthquake loading is in the order of the Poisson's ratio, the elastic modulus, the cohesive force, and the internal friction angle. The research findings of this paper provide scientific support for earthquake-resistance and disaster-reduction measures for underground metro tunnel engineering sites with overlying soft soil layers which are both sustainable and practical. However, due to limitations of space, the indoor vibration table test could not be further explored in relation to this engineering background. This study has significant theoretical and practical significance in the field of sustainability.

**Author Contributions:** Conceptualization, F.M. and S.L.; Methodology, Q.W.; Software, Y.N.; Formal analysis, Q.W.; Investigation, L.Z., Y.N., S.L. and Z.F.; Writing—original draft, Q.W.; Funding acquisition, F.M. All authors have read and agreed to the published version of the manuscript.

**Funding:** National Natural Science Foundation of China (51474045).

**Institutional Review Board Statement:** Not applicable.

**Informed Consent Statement:** Not applicable.

**Data Availability Statement:** Not applicable.

**Conflicts of Interest:** The authors declare no conflict of interest.

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
