# Peer review of "Study on the Surface Settlement of an Overlying Soft Soil Layer under the Action of an Earthquake at a Subway Tunnel Engineering Site"

_sustainability, doi:10.3390/su15129484_

Round 1
Reviewer 1 Report
The manuscript “Study on Surface Settlement of Subway Tunnel Engineering Site Overlying Soft Soil Layer under the Action of Earthquake” by Qiongyi Wang et al. is mainly conducted from the perspectives of statics and dynamics to study the characteristics of soft soils and the response of metro tunnels passing through soft soil layers. However, before the Editor decides, I suggest that the authors must consider that:
- The author should explain clearly in the abstract the novelty and the added value of this manuscript;
- References are suggestive. Add the DOI of the articles from References. I am convinced that it is useful for the manuscript if it will be included in the References section following papers with the same topics or using similar procedures, ex.:
- Mutual influence of geometric parameters and mechanical properties on thermal stresses in composite laminated plates with rectangular holes, Mathematics 2021, 9(4), 311; https://doi.org/10.3390/math904031;
- Study on Seismic Response and Vibration Reduction of Shield Tunnel Lining in Coastal Areas, Sustainability 2023, 15(5), 4185; https://doi.org/10.3390/su15054185
- Mechanics of Elastic Composites, Chapman & Hall/ CRC Press, U.S.A, 708 pp., (2003);
- Experimental Study of Dynamic Responses of Special Tunnel Sections under Near-Fault Ground Motio, Sustainability 2023, 15(5), 4506; https://doi.org/10.3390/su15054506.
For these reasons, I recommend the acceptance of this manuscript for publication after Minor Revision.
Author Response
Dear Dr,
Thank you very much for taking the time to review our manuscript entitled " Study on Surface Settlement of Subway Tunnel Engineering Site overlying Soft Soil Layer under the Action of Earthquake" We appreciate your valuable comments and suggestions, which have helped us to improve the quality of our work.
We have carefully considered all of your comments and suggestions, and have made the following revisions to the manuscript:
- Supplemented and revised.
- Supplemented.
We hope that these revisions address your concerns and have strengthened the manuscript. We have also attached a marked-up version of the revised manuscript to this email to show you exactly what changes have been made.
Once again, we appreciate your insightful feedback and the time you have dedicated to reviewing our work. Please let us know if you have any further questions or concerns.
Sincerely,
Qiongyi Wang

Reviewer 2 Report
This is a numerical study on the surface settlement of a tunnel on soft soil affected by earthquakes. ABAQUS software was used for the modelling and analysis of the tunnel and the soil under El-Centro quake waves. The study looks interesting and the authors provide a satisfactory level of information on the modelling and the results outputs. However, there are some spelling errors in the manuscript. The document needs some polishing possibly by professionals. Provided the following comments are addressed and after some additional polishing, the manuscript merits to be published in the Sustainability journal.
Line 102: What was the logic behind choosing the El-Cebtro wave rather than any other wave in this study?
Line 141: Please mention the source of the physical and mechanical parameters of the soil layers.
Line 164: Please mention the source for the local damping coefficient of 0.1491.
Line 170: Please provide more information regarding the selection of the mesh and their sizes.
Line 243: Please improve the quality of figures 6 - 15, if possible.
Needs polishing! See line 21 among other lines and the use of "trough" rather than "through", for example.
Author Response
Dear Dr,
Thank you very much for taking the time to review our manuscript entitled " Study on Surface Settlement of Subway Tunnel Engineering Site overlying Soft Soil Layer under the Action of Earthquake" We appreciate your valuable comments and suggestions, which have helped us to improve the quality of our work.
We have carefully considered all of your comments and suggestions, and have made the following revisions to the manuscript:
- The research presented in this article is based on the engineering background of a type 2 site. According to the "Code for Seismic Design of Buildings GB 50011-2010", the EL Centro wave data used in this article was compared with the response design spectrum of the type 2 site on the same chart (as shown in Figure 1). It can be seen that the EL Centro wave used in this article has a good fit with the type 2 site, indicating that it can be used for soil analysis in type 2 site.
- The parameters mentioned in line 154 are derived from the survey report and design data of the Dalian Metro Line 5 project on which this study is based.
- Reference: Lu, S.S.; Zhao, D.X.; Bai, J.K.; Liu S.D.; Yin, H. Vibration table test and numerical simulation study on the interaction system of tunnel-soil-bridge pile [J/OL]. Journal of Vibration Engineering: 1-15 [2023-05-11].
- Supplemented and revised.
- Optimized.
We hope that these revisions address your concerns and have strengthened the manuscript. We have also attached a marked-up version of the revised manuscript to this email to show you exactly what changes have been made.
Once again, we appreciate your insightful feedback and the time you have dedicated to reviewing our work. Please let us know if you have any further questions or concerns.
Sincerely,
Qiongyi Wang

Reviewer 3 Report
The manuscript titled "Study on Surface Settlement of Subway Tunnel Engineering Site overlying Soft Soil Layer under the Action of Earthquake" was carried out by using ABAQUS finite element analysis software. The findings of the study are important in terms of determining the behavior of such structures against possible earthquake activity and taking precautions against possible earthquake damage. Also, this study will contribute to the literature. I think that the manuscript is well written and organized. Therefore, it can be accepted for publication at "Sustainibility".
Author Response
Dear Dr,
Thank you very much for taking the time to review our manuscript entitled " Study on Surface Settlement of Subway Tunnel Engineering Site overlying Soft Soil Layer under the Action of Earthquake" We appreciate your valuable comments and suggestions, which have helped us to improve the quality of our work.
Thank you for your valuable comments on the review. We will continue to strive for improvement.
We hope that these revisions address your concerns and have strengthened the manuscript. We have also attached a marked-up version of the revised manuscript to this email to show you exactly what changes have been made.
Once again, we appreciate your insightful feedback and the time you have dedicated to reviewing our work. Please let us know if you have any further questions or concerns.
Sincerely,
Qiongyi Wang

Reviewer 4 Report
Review
This document studies the surface settlement of subway tunnel engineering site overlying soft soil layer under the action of earthquake. It is an outstanding paper, nevertheless, some suggestions are reported below.
1. The first paragraph is too long. Please rephase.
2. Add recent references supporting the idea of the introduction first paragraph.
3. Line 100 “…as the research object” replace with “as case of study”.
4. Line 197 replace with “Results and discussion.”
5. Please avoid the use of “we” in the text. Use indirect sentences.
6. In Figure 4 specify the units of the Y axis
7. Start the conclusion section summarizing the objectives of the manuscript.
8. In the conclusion section please highlight the limitations of the present manuscript and make some recommendations for future studies.
Some comments are found above for example:
1. The first paragraph is too long. Please rephase.
5. Please avoid the use of “we” in the text. Use indirect sentences.
Author Response
Dear Dr,
Thank you very much for taking the time to review our manuscript entitled " Study on Surface Settlement of Subway Tunnel Engineering Site overlying Soft Soil Layer under the Action of Earthquake" We appreciate your valuable comments and suggestions, which have helped us to improve the quality of our work.
We have carefully considered all of your comments and suggestions, and have made the following revisions to the manuscript:
- Rvised
- Supplemented
- Rvised
- Rvised
- Rvised
- The unit of the Y-axis in the horizontal direction is meter (m).
- Supplemented
- Supplemented
We hope that these revisions address your concerns and have strengthened the manuscript. We have also attached a marked-up version of the revised manuscript to this email to show you exactly what changes have been made.
Once again, we appreciate your insightful feedback and the time you have dedicated to reviewing our work. Please let us know if you have any further questions or concerns.
Sincerely,
Qiongyi Wang

Reviewer 5 Report
- Is this model in general or in a specific area? Please add that information to the title if it is based on a particular site.
- The abstract is too short and does not satisfactorily explain why this research is necessary; it needs more detailed information on the research background. In many instances, it is not required to read the complete text. From the abstract, the authors must give the essence of their report, preferably if they can follow the IMRaD structure since the readers extract the most valuable information from these. For example, why specifically on the subway tunnel on the Dalian Metro Line 5? Is there no same study before? Why three different seismic waves? How about the historical earthquake in this area? Also, the result needs more quantitative information (with exact values based on the model, uncertainty, etc.) rather than qualitative information.
- L39-43; L58-60: Are there any references to support this statement?
- L60-94: please summarize it, compare each other, and only show the most exciting and essential findings rather than explain their research one by one in a long paragraph. Based on the summary, the authors should put their research in a specific position where they will contribute to the topics.
- L97-99: "few studies"; please elaborate.
- The introduction section is too general and does not explicitly explain the urgency of this study in the Dalian Metro Line 5. The authors should explain in more detail why they chose a specific area, the research gap in the study area, how the current conditions related to the topics in the study area, etc.
- Since the readers of this article are an international community, the authors should give spatial information (a map) on the location they mentioned in the introduction and site conditions, such as the Dalian Metro Line 5, etc.
- Please elaborate on how the authors got the value in Tables 1 and 2.
- Since this is a model, the authors should explain in detail why they used those specific parameters and/or dimensions and which references they used to build that model. So far, we cannot see that detail and specific information clearly in the methods section. Furthermore, the most important thing, if we talk about the model, there is uncertainty and error. The authors did not give this information on how they validate their model and how they know whether the model is accurate or if there is uncertainty.
- If possible, please add a chart/diagram to summarize your (revised) methods from the start (data, input, etc.) to the end/output, including the (possible) problem that may exist during the simulation.
- Fig. 4, Please explain the meaning of each shape/color and information for the Y-axis.
- L262-263: Please avoid a paragraph that contains only one sentence.
- L356-383: the paragraphs are too long and sometimes hard to follow. The authors may divide into 2-3 paragraphs.
- The results part is too long, and somehow it is just a repetition of the illustration in the figures/table. It will be better if the authors can make it simple and efficient with not too much text. Also, it would be better if they could separate the Results and Discussion parts. Furthermore, in this version, the manuscript has no real discussion, and where interpretations are expected, it lacks detailed referencing to demonstrate that the author tries to contextualize their results. It is sad that the author's work could be better presented, supported, and discussed contribution. The manuscript needs a proper, scientifically correct discussion and more in-depth arguments, including model validation, accuracy, and uncertainty (if any).
- Conclusions: in general, conclusions are too long and not so easy to read. It also confirms that the identification of the main contribution of the manuscript is unclear.
Author Response
Dear Dr,
Thank you very much for taking the time to review our manuscript entitled " Study on Surface Settlement of Subway Tunnel Engineering Site overlying Soft Soil Layer under the Action of Earthquake" We appreciate your valuable comments and suggestions, which have helped us to improve the quality of our work.
We have carefully considered all of your comments and suggestions, and have made the following revisions to the manuscript:
- This article focuses on the study of the surface settlement characteristics of the subway tunnel in the overlying soft soil layer under seismic action, with the Dalian Metro Line 5 as the research background. It can be considered as a general study.
- The significance of the research in the abstract has been supplemented. The authors of this paper participated in the construction of Dalian Metro Line 5 and found a scientific problem that needs to be solved in this engineering context: what is the settlement law of the surface above the soft soil layer in the subway tunnel under earthquake action? No research on this issue was found in the relevant literature. The selection of the three types of seismic waves was based on the seismic design code "Code for Seismic Design of Buildings GB 50011-2010", which has clear seismic requirements for engineering construction in the region.
- Based on the distribution map of Chinese soft soil and the map of subway line construction, it can be seen that L39-43 is the study area. According to the Chinese Seismic Ground Motion Parameter Zonation Map (GB18306-2015), the seismic activity characteristics of the study area are L58-60.
- It has already been summarized in the text.
- There is no research on the surface settlement law of the overlying soft soil layer of the subway tunnel under seismic action found in the literature review.
- The author of this article participated in the construction of Dalian Metro Line 5 and found a scientific problem that needs to be solved in this engineering context - what is the settlement law of the ground surface at the top of the subway tunnel in the overlying soft soil layer under earthquake action. However, no research on this issue was found in the relevant literature, so this scientific problem was studied based on Dalian Metro Line 5.
- Supplemented
- The source of the parameters is explained in line 154, which is based on the survey report and design documents of the Dalian Metro Line 5 project.
- The reference for the establishment of this model is: Lu, S.S.; Zhao, D.X.; Bai, J.K.; Liu S.D.; Yin, H. Vibration table test and numerical simulation study on the interaction system of tunnel-soil-bridge pile [J/OL]. Journal of Vibration Engineering: 1-15 [2023-05-11].
- Due to space limitations, specific simulation scenarios were not provided.
- The shape is in the form of a grid partition, and the colors represent different levels of settlement deformation. The horizontal direction is the Y-axis, and the unit is in meters.
- Rvised
- Rvised
- Rvised. Model establishment reference: Lu, S.S.; Zhao, D.X.; Bai, J.K.; Liu S.D.; Yin, H. Vibration table test and numerical simulation study on the interaction system of tunnel-soil-bridge pile [J/OL]. Journal of Vibration Engineering: 1-15 [2023-05-11].
- Rvised
We hope that these revisions address your concerns and have strengthened the manuscript. We have also attached a marked-up version of the revised manuscript to this email to show you exactly what changes have been made.
Once again, we appreciate your insightful feedback and the time you have dedicated to reviewing our work. Please let us know if you have any further questions or concerns.
Sincerely,
Qiongyi Wang

Round 2
Reviewer 5 Report
Based on the revised manuscript, I still firmly believe that the manuscript is not acceptable yet for the Sustainability journal. The improvements made by the authors are not profound and insufficient to make this paper a scientific contribution. The map is not using good and well-cartographic rules, i.e., coordinates, scale, inset, legends, etc. Furthermore, the most important thing is the manuscript still has no real discussion yet (we cannot see a proper and scientifically correct discussion), and also, where interpretations are expected, it lacks detailed referencing to demonstrate that the author is trying to put their results in context. I am concerned that the results of this study will not contribute to research and practice.
Author Response
Dear Dr,
Thank you very much for reviewing our paper and providing valuable comments and suggestions. We have carefully read your review and have given careful consideration to the issues you raised. Below is our specific response to your review comments:
1.This article focuses on the study of the surface settlement characteristics of the subway tunnel in the overlying soft soil layer under seismic action, with the Dalian Metro Line 5 as the research background. It can be considered as a general study.
2.The significance of the study has been supplemented in the abstract. The authors of this paper were involved in the construction of Dalian Metro Line 5 and identified a scientific problem that needs to be addressed in the context of this project, which is the settlement behavior of the top surface of the subway tunnel in the overlying soft soil layer under seismic loading. Upon reviewing the relevant literature, no studies on this specific issue in the region were found.
The selection of three seismic waves is based on the seismic design code "Code for Seismic Design of Buildings GB 50011-2010" and "Seismic Ground Motion Parameters Zoning Map of China GB18306-2015". Dalian City is located in Seismic Intensity Grade VII, and for seismic design of underground structures, it is recommended to consider three different seismic waves that are approved by national regulations as the seismic intensity basis for regional seismic design. In general, the seismic intensity with a 10% exceedance probability within a 50-year period is chosen. Therefore, this paper selects artificial seismic waves with a 10% exceedance probability as one of the seismic waves.
The selected El-Centro wave is mainly derived from the 1940 Imperial Valley earthquake in the United States, and the seismic wave data are collected from different seismic monitoring stations. The El-Centro wave data are designed to have good agreement with the response spectrum of Category II site response design. The selected Kobe wave primarily considers the strong deformation of the site soil during an earthquake process.
3.Based on the distribution map of Chinese soft soil and the map of subway line construction, it can be seen that L39-43 is the study area. According to the Chinese Seismic Ground Motion Parameter Zonation Map (GB18306-2015), the seismic activity characteristics of the study area are L58-60.
4.A summary was provided in L97-101.
5.There is a scarcity of research in the literature regarding the settlement patterns of the surface above subway tunnels' overlying soft soil layer under seismic conditions.
6.Added information: According to the "Seismic Ground Motion Parameter Zoning Map of China GB18306-2015," Dalian City is located in the Seismic Intensity Zone VII. This makes it crucial to study the seismic response of underground engineering and important structures in the region. The authors of this paper were involved in the construction of Dalian Metro Line 5 and identified a scientific problem that needs to be addressed in this engineering context. Specifically, the research aims to investigate the settlement patterns of the surface above the subway tunnel's overlying soft soil layer under seismic conditions. Upon reviewing the relevant literature, no previous studies on this specific issue in the region were found. Therefore, the research is conducted based on the Dalian Metro Line 5 project to address this scientific problem in the area.
7.Supplemented.
8.The source of the parameters is explained in line 154, which is based on the survey report and design documents of the Dalian Metro Line 5 project.
9.Reference for model establishment: Lu, S., Zhao, D., Bai, J., Liu, S., Yin, H. Vibration table test and numerical simulation of tunnel-soil-bridge pile interaction system. Journal of Vibration Engineering, 1-15 [2023-05-11]. EI. This reference is classified as a top journal in the field of vibration in China. Furthermore, the authors of this paper and the referenced paper belong to the same research group, and the methodology for model establishment is consistent.
10.Due to space limitations, specific simulation scenarios were not provided.
11.The shape is in the form of a grid partition, and the colors represent different levels of settlement deformation. The horizontal direction is the Y-axis, and the unit is in meters.
12.Rvised.
13.Rvised.
14.The reference for establishing the model has been added: Lu, S., Zhao, D., Bai, J., Liu, S.,& Yin, H.(2023. Vibration table test and numerical simulation of tunnel-soil-bridge pile interaction system. Journal of Vibration Engineering,1-15. [Online]. Retrieved May 11,2023, from EI. This reference is from a top seismic journal in China, and the authors of this paper and the reference are from the same research group, using the same method for model establishment.
15.The conclusions 1 and 3 have been modified, and a compass rose, scale, and legend have been added to the figures.
Once again, we sincerely thank you for your review comments and suggestions, which provide important guidance and directions for improving our research. We will carefully consider your comments and make corresponding modifications and adjustments in the final paper. If you have any further questions or suggestions, we are more than willing to listen and respond. Thank you for your time and support!
Sincerely,
Qiongyi Wang
